# Epidemiology Meets Advocacy: Understanding Pediatric Dental Trauma and Delayed Care in Post-Conflict Syria

**DOI:** 10.3390/ijerph22121864

**Published:** 2025-12-15

**Authors:** Yasser Alsayed Tolibah, Nada Bshara, Ramah E. Makieh, Marwan Alhaji, Mohammed N. Al-Shiekh, MHD Bashier AlMonakel, Osama Aljabban, Ziad D. Baghdadi

**Affiliations:** 1Department of Pediatric Dentistry, Faculty of Dentistry, Damascus University, Damascus P.O. Box 3062, Syria; yasseralsayedtolibah@gmail.com (Y.A.T.); gmmn2012@gmail.com (N.B.); rmakkieh@yahoo.com (R.E.M.); mrwan.2013.55.a@gmail.com (M.A.); shiekh.mon@gmail.com (M.N.A.-S.); mohamadbashier@hotmail.com (M.B.A.); 2Department of Endodontics, Damascus University, Damascus P.O. Box 3062, Syria; dr.ossamaljabban@gmail.com; 3Department of Preventive Dental Sciences, Division of Pediatric Dentistry, University of Manitoba, Winnipeg, MB R3E 0W2, Canada; 4Centre for Community Oral Health, Dr. Gerald Niznick College of Dentistry, University of Manitoba, P131B, 780 Bannatyne Avenue, Winnipeg, MB R3E 0W2, Canada; 5TopSmiles Pediatric Dentistry & Orthodontics, Winnipeg, MB R2M 3A4, Canada

**Keywords:** traumatic dental injuries, prevalence, risk factors, case presentations, Syria, pediatric dentistry

## Abstract

**Highlights:**

**Public health relevance—How does this work relate to a public health issue?**
Traumatic dental injuries (TDIs) impose a substantial burden on children’s well-being, contributing to pain, functional limitations, aesthetic concerns, and long-term treatment needs—all of which can adversely affect daily functioning, quality of life, and school performance.The high proportion of injuries occurring at home and the substantial delays in seeking care highlight gaps in caregiver awareness, environmental safety, and access to timely emergency dental services. These factors reflect fundamental components of child health systems and emergency preparedness.

**Public health significance—Why is this work of significance to public health?**
This study presents the first recent, large-scale institutional data on TDIs among Syrian children during a period of national recovery, providing essential epidemiological evidence to inform preventive strategies and emergency care planning.Identification of key risk groups—males, younger children, children with special healthcare needs, and those with a history of previous trauma—supports the development of targeted interventions aimed at reducing oral health disparities and preventing avoidable complications such as pulp disease and tooth loss.

**Public health implications—What are the key implications or messages for practitioners, policy makers and/or researchers in public health?**
**For practitioners:** Early recognition, prompt referral, and consistent adherence to evidence-based trauma management protocols are critical to preventing pulp necrosis and apical pathology—particularly in settings where delayed presentation is common.**For policymakers and researchers:** The findings underscore the need to establish national TDI surveillance systems, integrate dental trauma first-aid education into school health curricula, and develop family-centered preventive strategies for home environments. Special emphasis is warranted for vulnerable groups, such as children with special healthcare needs, who experience higher trauma burdens.

**Abstract:**

Objective. To evaluate the prevalence, risk factors, aetiology, and management of traumatic dental injuries (TDIs) among children aged 1–18 years attending the Department of Pediatric Dentistry, Damascus University, Syria, during 2023–2024, and to illustrate representative clinical cases with documented outcomes. Methods. This retrospective cross-sectional study reviewed 2716 patient records (2023–2024) and identified 301 children with TDIs. Demographic, clinical, and behavioural variables were extracted and analysed using χ^2^, *t* tests, ANOVA, and binary logistic regression (IBM SPSS v26). Results. The overall TDI prevalence was 11.08%. Males were over twice as likely as females to experience TDIs (OR = 2.30; 95% CI = 1.76–3.01; *p* < 0.001). Older age acted as a protective factor (OR = 0.56; 95% CI = 0.43–0.74; *p* < 0.001). Falls were the most common cause (63.7%), and injuries most often occurred at home (48.9%). The maxillary central incisors were most frequently affected (68.5% of cases). Children with special healthcare needs had significantly more traumatised teeth (mean = 2.61 ± 1.13) than healthy children (1.66 ± 0.92; *p* < 0.001). Nearly half of the patients (45.3%) presented > one month after injury, and asymptomatic apical periodontitis and reversible pulpitis were the most frequent diagnoses. Representative case presentations demonstrated multidisciplinary management using restorative, endodontic, and orthodontic approaches with favourable follow-up outcomes. Conclusions. TDIs affected about one in nine children in this Syrian cohort. Male gender, younger age, and previous trauma were key risk factors. The predominance of delayed presentation underscores the need for community education, early referral systems, and targeted preventive programs within school and home environments.

## 1. Introduction

Traumatic dental injuries (TDIs) represent a major public health concern worldwide, particularly among children and adolescents. They rank second only to dental caries in prevalence, preceding periodontal disease as a cause of oral morbidity and functional impairment [1,2,3]. Beyond their physical consequences, TDIs often result in significant psychological distress, aesthetic concerns, and economic burden for affected families, collectively diminishing children’s oral health-related quality of life [4,5,6].

Epidemiological studies indicate a wide variation in the global prevalence of TDIs, ranging from 4% to over 40%, depending on population characteristics, study design, diagnostic criteria, and sociocultural context [7,8,9]. A meta-analysis by Petti et al. [3] estimated that more than one billion people have experienced a TDI, with prevalence rates of 22.7% in primary dentition and 15.2% in permanent dentition. Such disparities across regions are influenced by behavioural, environmental, and systemic factors—including sports participation, unsafe environments, and socioeconomic inequality.

Risk factors for TDIs are multifactorial. Established determinants include male gender, increased overjet, inadequate lip coverage, history of previous trauma, and participation in contact sports [10,11,12,13]. Glendor [14] grouped these into three broad categories: (1) oral factors, such as dental malocclusion and overjet; (2) environmental factors, including material deprivation and unsafe surroundings; and (3) behavioural factors, such as aggression and exposure to violence. More recent literature has expanded this framework to include systemic conditions, developmental disorders, and sensory or motor impairments that increase susceptibility to trauma [15]. Children with special healthcare needs are especially vulnerable due to impaired motor coordination, limited protective reflexes, and behavioural challenges [16].

Favourable outcomes following TDIs depend on public awareness of emergency management and timely access to professional care [17,18]. Since prognosis worsens with delayed treatment, preventive education and organized emergency dental services are essential components of child oral health policy [19].

### 1.1. Local Context

In Syria, published data on traumatic dental injuries (TDIs) remain scarce and fragmented despite the increasing relevance of this public health problem. Over two decades ago, Marcenes et al. [20] reported relatively low TDI prevalence among Syrian schoolchildren (5.2% at age 9 and 11.7% at age 12). More recent data suggest a rise in dental trauma prevalence among Syrian children. For example, Alshayeb et al. [21] reported a prevalence of 17.6% among 9–12-year-olds and discussed several contextual factors—such as increased exposure to injury hazards and limited access to dental care—that may have contributed to these findings. Other studies [22,23,24] conducted during the conflict period have described environmental and healthcare system challenges, including overcrowded living conditions, reduced access to safe recreational areas, and constrained dental service availability. While these contextual conditions have been cited as potential contributors to injury risk, they were not directly assessed in the present study and should be interpreted cautiously. These associations reflect interpretations in the cited studies and should be considered cautiously, given the limitations of the available epidemiological data. Recent publications have reported shifts in environmental and healthcare conditions that may influence injury risk—for example, overcrowded living spaces, limited availability of safe play areas, and disruptions in access to dental services. These contextual factors have been cited as potential contributors to changes in TDI epidemiology, but should be interpreted within the limits of currently available data [22].

Beyond community prevalence, recent Syrian studies highlight additional domains where traumatic dental injuries constitute an overlooked burden. A national survey among anesthetists reported that 38.8% had caused dental trauma during their professional practice, yet most lacked adequate knowledge or training in TDI classification, prevention, and emergency management. Notably, 57% acknowledged insufficient understanding of oral anatomy and dental injury protocols, underscoring systemic gaps in healthcare providers’ preparedness to manage TDIs during peri-anesthetic procedures. This reflects a broader deficiency in trauma-awareness education across medical disciplines, especially under crisis conditions [23].

Similarly, a recent investigation involving 582 young athletes in Damascus revealed a high prevalence of orofacial-dental trauma (45.5%), with injury patterns strongly influenced by age, sport type, and training exposure. Protective gear use remained strikingly low (27.3%), and falls during training accounted for the leading mechanism of injury. These findings emphasize that TDIs in Syria extend beyond the general pediatric population to include high-risk groups such as athletes, whose injuries are exacerbated by inadequate preventive measures and limited safety regulations [24].

Collectively, published studies suggest that multiple contextual factors—including environmental hazards, limited provider training, and reduced preventive awareness—may influence TDI occurrence in various Syrian subpopulations. However, the specific contribution of each factor requires further investigation.

It is important to note that the present study did not directly measure conflict-related exposures; references to contextual conditions are included solely to summarize previously reported factors that may inform hypothesis generation and interpretation.

### 1.2. Rationale and Aim

Despite the well-established global determinants of traumatic dental injuries, the extent to which these factors operate within the Syrian context remains poorly understood. Existing national evidence is limited to a few isolated studies, many of which were conducted before the humanitarian crisis or based on small or non-representative samples. The prolonged conflict has created unique risk conditions—such as widespread environmental hazards, increased exposure to violence, and reduced access to preventive and emergency dental care—that have not been adequately examined in recent epidemiological research. This gap highlights the urgent need for updated institutional data capable of reflecting the post-conflict realities affecting Syrian children. By analyzing cases presented to Damascus University, a major referral center receiving patients from diverse regions, the present study aims to provide contemporary evidence that directly addresses this national data deficit.

The Department of Pediatric Dentistry at Damascus University serves as a national referral center providing comprehensive dental care to children and individuals with special needs. Understanding the prevalence, etiology, and management of TDIs in this context is crucial for informing both academic curricula and public health initiatives. While the study was conducted in an institutional, urban setting (Damascus University), the diversity of patients referred from multiple governorates supports cautious generalization beyond the Damascus population.

### 1.3. Hypothesis and Research Question

This study aimed to document the prevalence, risk factors, and treatment characteristics of TDIs among children attending Damascus University between 2023 and 2024, with selected case presentations illustrating typical clinical scenarios.

We hypothesized that the prevalence and patterns of TDIs in this clinical setting would vary significantly by age, gender, special healthcare needs (SHCN) status, and the cause and anatomic location of injury. Accordingly, the primary research question was: Do demographic and clinical factors—including age, sex, SHCN status, trauma etiology, and injury location—significantly influence the presentation and treatment characteristics of TDIs among children treated in a post-conflict Syrian context?

## 2. Materials and Methods

### 2.1. Study Design and Ethical Approval

This retrospective cross-sectional study was conducted at the Department of Pediatric Dentistry, Faculty of Dentistry, Damascus University, Syria. It aimed to determine the prevalence, risk factors, and management patterns of traumatic dental injuries (TDIs) among children aged 1–18 years who attended the department between January 2023 and December 2024.

The study followed the Preferred Reporting Items for Observational Studies in Endodontics (PROBE 2023) guidelines [25]. Ethical approval was obtained from the Research Ethics Committee of Damascus University (Approval No. 2654/13.02.2023) and from the Ministry of Higher Education and Research. The research adhered to the principles of the Declaration of Helsinki (2013 revision). Caregivers of all patients had provided written informed consent at the time of initial treatment, allowing anonymized use of patient records for academic and research purposes.

### 2.2. Study Population and Sample Selection

The study population consisted of all pediatric patients (n = 2990) who received treatment at the Department of Pediatric Dentistry during the 2023–2024 period. The department functions as a tertiary public referral center, treating children from Damascus and surrounding governorates. All available patient records were reviewed.

Inclusion criteria were as follows:Children aged 1–18 years;Diagnosed and treated for dental trauma during the study period;Complete documentation of demographic, diagnostic, and treatment data.

To clarify the source and nature of included trauma cases, patients were enrolled only when the TDI itself was the reason for presentation to the pediatric dental clinic or was identified as part of an oral examination following referral after a general bodily trauma. However, cases of old or previously healed trauma that were incidentally detected during routine dental visits and not related to the presenting complaint were not included (estimated as “Previous history of trauma”), unless the patient sought care specifically for complications directly attributable to that prior trauma. Thus, the study sample reflects children who required clinical assessment or treatment for an active or clinically relevant TDI episode during the study period.

Records with missing or illegible essential information (demographics, trauma diagnosis, or treatment details) were excluded (n = 64, ≈2%). The final analytical sample consisted of 2716 valid records, including 301 TDI cases (11.08%) and 2415 non-trauma cases (control group). The control group, matched by age and gender, consisted of children who attended the same clinic during the same period for other dental conditions (caries, malocclusion, and preventive care) and were used for comparative statistical analysis. The control group was matched to the TDI group by age and sex using frequency matching, ensuring that the distributions of these variables in the control sample matched those observed in children with TDIs. Individual one-to-one pairing was not performed. Frequency matching was selected to preserve the full sample size of eligible non-trauma records while maintaining comparability across key demographic variables. This matching strategy supports reproducibility and aligns with the analytical framework used in subsequent comparative analyses.

### 2.3. Data Extraction and Calibration

Two trained examiners (Y.A.T. and M.A.) independently screened and extracted data from the records using a standardized form. Before data collection, both underwent a calibration process supervised by an experienced pediatric dentist (N.B.) to ensure diagnostic consistency. A pilot evaluation of 30 randomly selected records was conducted independently by both examiners, and discrepancies were resolved through discussion until full agreement was reached.

The inter-examiner reliability was assessed using Cohen’s kappa coefficient (κ = 0.89), indicating excellent agreement. Continuous supervision and periodic cross-checking were maintained throughout data collection to minimize observer bias.

The calibration process covered TDI classification, apex status assessment, and occlusal variables, but not the behavioral assessment.

### 2.4. Variables and Diagnostic Criteria

The following data were extracted:
Demographic variables: age, gender, and governorate of residence.Health status: classified as healthy or special healthcare needs (SHCN). SHCN categories followed Almonaqel’s classification [26]: neurological disorders, sensory impairments, behavioural disabilities, medical disabilities, and developmental/congenital impairments. SHCN status was confirmed through medical documentation provided by caregivers and verified by the supervising pediatric dentist.Behavioural assessment: rated according to the Frankl Behaviour Rating Scale (definitely negative, negative, positive, definitely positive).Occlusion type: assessed using Angle’s classification and primary molar relationships (flush terminal plane, distal step, mesial step).Caries experience: recorded as dmft/DMFT indices following WHO criteria [27].Previous history of trauma: yes/no, based on caregiver report and previous record review.Apex status: classified radiographically as open or closed, according to Gill et al. [28].Etiological and situational variables: time between trauma and dental visit in weeks (Delayed presentation was defined as a time interval >4 weeks between the date of traumatic injury and the first dental visit), place of injury (home, school, street, etc.) [29,30], and cause (falls, traffic accidents, collisions, fights).

TDIs were classified according to the 2022 World Health Organization Classification of Traumatic Dental Injuries (NA0D) [31]. This classification system was applied from the outset during the completion of the standardized trauma examination form, ensuring that all patient records were documented consistently under the same diagnostic criteria. The WHO classification includes two major categories:Fracture-related injuries (enamel fracture, enamel–dentin fracture, complicated crown fracture, crown–root fracture, root fracture).Luxation-related injuries (concussion, subluxation, extrusive, lateral, and intrusive luxation, avulsion).

### 2.5. Data Management and Quality Control

All data were entered into Microsoft Excel 2021 and cross-checked for accuracy by an independent reviewer (R.M.). Each child was assigned a unique identification number to prevent duplication of cases or overcounting of multiple visits related to the same traumatic event. Only the first visit for each trauma episode was included; re-treatments and follow-ups were excluded.

Incomplete or unclear entries were flagged and discussed among the investigators. Random audits were performed to verify data integrity and minimize transcription errors.

### 2.6. Statistical Analysis

Data were analyzed using IBM SPSS Statistics version 26 (IBM Corp., Armonk, NY, USA). Descriptive statistics (frequencies, means, and standard deviations) were computed for all variables. The normality of continuous data distributions was assessed using the Kolmogorov–Smirnov test, while the Levene test was applied to evaluate the homogeneity of variances before conducting parametric analyses.

Inferential analyses included the following:Chi-square tests (χ^2^) for categorical variable comparisons.Independent-samples *t* tests and ANOVA with Bonferroni post hoc for continuous variables.Binary logistic regression to identify independent predictors of TDIs (dependent variable: injury presence = yes/no).

Odds ratios (OR) with 95% confidence intervals (CI) were reported, and a *p*-value < 0.05 was considered statistically significant.

### 2.7. Methodological Considerations

Given the retrospective hospital-based design, findings reflect patterns among children seeking care at an academic dental institution and may not represent the full community prevalence of TDIs. However, the large sample size, rigorous calibration, and standardised classification enhance both internal validity and reproducibility.

## 3. Results

### 3.1. TDIs Prevalence and the Associated Risk Factors of the Sample

A total of 2716 patient records were analyzed, of which 301 children (11.08%) presented with traumatic dental injuries (TDIs)—the remaining 2415 served as the control group (Figure 1). Male children had a significantly higher prevalence (13.9%) compared with females (7.5%) (*p* < 0.001). The 12–18-year age group showed the highest rate (18.2%), compared with younger children (*p* < 0.001) (Table 1). Logistic regression identified male gender (OR = 2.30; 95% CI: 1.76–3.01) as a strong independent predictor, while increasing age (OR = 0.56; 95% CI: 0.43–0.74) was protective (Table 2).

To aid clinical interpretation, effect sizes were reported using odds ratios (ORs) and 95% confidence intervals. Notably, male gender (OR = 2.30) and younger age (OR = 0.56) remained strong independent predictors even after adjusting for other variables.

Although the logistic regression model was statistically significant, the explained variance was modest (Nagelkerke R^2^ = 0.118), indicating that a substantial proportion of the variability in TDI occurrence remains unexplained. This suggests the presence of unmeasured confounding factors, such as environmental hazards, parental supervision, socioeconomic status, and activity level, which were not captured in the current dataset but may strongly contribute to trauma risk.

The Frankl behavioural scale showed a significant association with traumatic dental injuries in univariate analyses, but this significance disappeared in the multivariable logistic regression model. This attenuation may be explained by collinearity with age and gender, since behavioural cooperation in younger children typically overlaps with the developmental stage rather than representing an independent predictor. Once age and gender were included in the model, the behavioural effect was substantially reduced.

### 3.2. Caries Indices and Oral Health Correlates

The mean caries indices (DMFT/dmft) were significantly higher in non-trauma cases than in trauma cases (*p* < 0.001), suggesting that caries experience was not a risk factor for dental trauma. Children with TDIs generally showed good oral hygiene and predominantly exhibited positive or definitely positive behaviour, according to the Frankl scale (Table 3).

### 3.3. Injury Characteristics and Treatment

Table 4 summarizes the distribution of trauma type, pulp diagnosis, time to presentation, and treatment modality. Complicated crown fractures (39.1%) were the most frequent injuries among permanent teeth, whereas complicated crown–root fractures (23.1%) predominated in primary teeth. Falls represented the leading cause (63.7%), and nearly half the children (45.3%) presented more than one month after the injury, underscoring the delay in seeking care. Asymptomatic apical periodontitis (32.1%) and reversible pulpitis (29.6%) were the most frequent pulp diagnoses in permanent teeth.

### 3.4. Distribution by Age, Health Status, and Etiology

The mean number of traumatized teeth per child was 1.75 ± 0.98, ranging from 1 to 5. SHCN children had a significantly higher mean (2.61 ± 1.13; *p* < 0.001). Children under 6 years old had the highest mean number of affected teeth, reflecting the greater vulnerability of the primary dentition.

Figure 2 illustrates the mean number of traumatized teeth by age group, and Figure 3 displays the proportional distribution of etiological causes.

It is noteworthy that most patients (74%) whose TDIs resulted from traffic accidents had received initial medical care at the hospital—typically involving suturing of lacerated oral tissues—before being referred to the Faculty of Dentistry for dental trauma management. In contrast, the majority of patients whose injuries were caused by non-traffic-related incidents presented directly to the Faculty of Dentistry for immediate dental trauma care.

### 3.5. Illustrative Clinical Cases

Representative clinical cases managed within the department are presented in Figure 3, Figure 4 and Figure 5, demonstrating orthodontic–prosthodontic, surgical, and regenerative approaches, respectively, with follow-ups confirming functional and aesthetic success. All case images were published with documented parental consent and anonymization.


**Case 1 (Figure 4):**


A 14-year-old adolescent presented to the Department of Pediatric Dentistry 18 months after a traumatic dental injury. The trauma had resulted in complete avulsion of tooth #21, accompanied by space loss due to the drift of teeth #11 and #22 into the edentulous area. The case was managed using a fixed orthodontic appliance to regain the lost space, followed by prosthetic rehabilitation with an adhesive bridge to replace the missing tooth. The overall treatment duration was seven months.


**Case 2 (Figure 5):**


A 7-year-old child presented to the Department of Pediatric Dentistry three days after the injury with a complicated crown fracture involving pulp exposure in tooth #42. Radiographic examination revealed that the fractured crown fragment was embedded within the lower lip. As the tooth had a closed apex, the fragment was surgically removed from the lip, followed by traditional endodontic treatment of tooth #42 in the same visit. The fractured crown segment was successfully reattached, restoring the tooth’s original form and function.


**Case 3 (Figure 6):**


A 9-year-old child presented to the Department of Pediatric Dentistry with a history of recurrent dental trauma. The first injury, one year prior, had resulted in an uncomplicated crown fracture of tooth #11, still untreated. In contrast, the second trauma, one month earlier, caused an uncomplicated crown fracture of tooth #21. The child’s main complaint was pain associated with tooth #21. Routine radiographic examination revealed pulp necrosis in tooth #11. Vital pulp therapy with MTA was performed on tooth #21, whereas tooth #11 was managed with an MTA apical plug.

A two-year follow-up radiograph showed a marked reduction in the apical radiolucency of tooth #11 and the formation of a mineralized apical barrier adjacent to the plug. In contrast, tooth #21 later developed symptoms of acute apical periodontitis, necessitating additional endodontic intervention.

### 3.6. TDI Risk Factors According to the Age Classification

In children with TDIs, a significant difference was found when comparing age classification and gender; both males and females in the 6–12-year age group were most frequently affected by TDIs. Although healthy children aged 6–12 years were the most commonly affected by TDIs, the incidence of TDIs among SHCN children was relatively evenly distributed across all age groups. No association was found between a previous traumatic injury and its recurrence in all age groups.

The highest percentages of traumas were due to falls and collisions with objects in the 6- to12 age group (43.3% and 9.0%, respectively). However, the highest percentage of trauma caused by fights was in the 12-year-old ≤ age group. Traffic accidents as a cause of dental trauma were similar across all age groups (Table 5).

### 3.7. TDIs’ Risk Factors According to the Number of Traumatized Teeth

The mean number of traumatized teeth was higher among SHCN children (2.61 ± 1.133) than among healthy children (1.66 ± 0.918), with a significant difference (*p* < 0.001). The highest mean number of traumatized teeth was in the age group under 6 years (reflecting the primary dentition stage), with significant differences. Finally, the number of traumatized teeth varied significantly by cause of TDIs. Traffic accidents resulted in the highest number of affected teeth, followed by collisions with objects (2.93 ± 1.944 and 2.04 ± 1.105, respectively), with statistically significant differences (Table 6).

Significant differences in the mean number of traumatized teeth were found between primary dentition and mixed dentition, as well as between primary dentition and permanent dentition (*p* < 0.05). The highest mean was observed in the under-6 age group (2.34 ± 1.234), with significant differences noted when post hoc tests were applied. The mean number of traumatized teeth due to traffic accidents was the highest among trauma causes, with significant differences (Table 7).

## 4. Discussion

In interpreting these findings, we distinguish between results that arise directly from our dataset and contextual explanations derived from previously published literature. The latter are included to support hypothesis generation and to situate the findings within broader environmental and systemic conditions, but are not measured variables in the present study.

### 4.1. Overview and Key Findings

This study provides updated epidemiological evidence on TDIs among children and adolescents in Syria—one of the few national-level datasets collected in a post-conflict setting. The overall prevalence of 11.08% aligns with global averages (10–15%) [3] and demonstrates the continuing public health burden of dental trauma in low- and middle-income contexts. Male gender, younger age, and a history of previous trauma were significant predictors of TDIs. Falls at home were the most common cause (63.7%), and our dataset shows that nearly half of patients (45.3%) presented more than one month after injury, and these late presentations were significantly associated with asymptomatic apical periodontitis. While the current study did not measure the underlying reasons for delayed care, previous literature has suggested that factors such as limited parental awareness, logistic constraints, and disrupted access to dental services may contribute to late presentation in comparable settings. These contextual explanations should therefore be interpreted as theoretical frameworks rather than findings derived from our data. Importantly, these findings are consistent with established time-dependent prognostic patterns described in the IADT 2020 guidelines, which emphasize that delayed professional management markedly increases the likelihood of pulp necrosis and subsequent apical pathology [32]. The observed distribution of pulp diagnoses in our cohort, particularly the high proportion of asymptomatic apical periodontitis among late presenters, aligns with these well-recognized biological responses to untreated trauma. Several barriers may contribute to this delay, including limited parental awareness of the urgency of trauma management, logistical and financial constraints associated with accessing specialized dental services during ongoing socioeconomic instability, and a lack of public knowledge that the Faculty of Dentistry offers dedicated trauma care.

Furthermore, the finding that children with SHCN sustained a higher number of traumatized teeth is noteworthy. This elevated risk may be explained by a combination of functional and behavioral factors frequently reported among SHCN populations, including impaired motor coordination, reduced protective reflexes, inadequate lip coverage or increased overjet, difficulties with balance and gait, and behavioral challenges such as impulsivity or lack of safe play supervision. These vulnerabilities increase the likelihood of falls, collisions, or unprotected injuries, particularly in unsupervised environments. Moreover, children with SHCN often face barriers in accessing timely dental care, which may exacerbate the severity of injuries and hinder prompt management. Such risk profiles are well documented in the literature—for instance, a recent systematic review found that overjet and lip incompetence significantly contribute to higher rates of traumatic dental injuries in individuals with SHCN. Therefore, our results likely reflect an interaction between the inherent functional limitations associated with SHCN and the heightened exposure to environmental and care-related risk factors in our setting. The present dataset demonstrates that children with SHCN experienced a significantly higher mean number of traumatized teeth. Although our study did not directly assess mechanisms underlying this difference, prior research suggests that factors such as impaired motor coordination, reduced protective reflexes, or behavioural challenges may increase susceptibility to trauma in SHCN populations. These explanations are therefore contextual and not measured determinants within our dataset.

### 4.2. Global Comparisons

The prevalence observed in this study is consistent with international reports: India (13%) [11], and Turkey (4.4%) [7], yet notably lower than Saudi Arabia (44%) [8] and Sweden (37.6%) [33] and some European urban cohorts where organized sports and outdoor activities increase exposure to injury [34].

Gender distribution (71% male, 29% female) mirrors results from Lebanon (male-to-female ratio 2.1:1) [12] and Jordan (2.3:1) [35], reaffirming that male children are more likely to experience trauma due to greater physical activity and risk-taking behaviour.

Age-related trends in the present study, with a peak between 6–12 years, also correspond with studies from Greece [36] and India [11], reflecting the developmental stage of mixed dentition when increased mobility and participation in sports coincide with incomplete motor coordination.

When comparing the time to presentation, the proportion of delayed visits (more than one month, 45.3%) was substantially higher than reported in Jordan (12.5%) [35] and Lebanon (17%) [12], highlighting critical barriers to timely trauma care in Syria. Delayed management likely contributed to the high rates of asymptomatic apical periodontitis (32.1%) and irreversible pulpitis (9%) observed, emphasizing the urgent need for public awareness campaigns on early intervention.

### 4.3. Sociopolitical and Environmental Context

The following contextual considerations are derived from external reports and are presented solely to help situate our findings within known environmental and healthcare constraints documented during the conflict period. These factors were not measured in our dataset and should not be interpreted as causal determinants within this study.

As mentioned, several studies have described disruptions to infrastructure, public safety, and healthcare access during the conflict period. Such contextual elements have been hypothesized to influence injury patterns, including dental trauma. While these reports provide a plausible backdrop for understanding the predominance of home-based injuries and delays in care observed in our cohort, they represent theoretical interpretations rather than empirical findings of the present study.

The results must be interpreted within the broader sociopolitical and environmental context of Syria. Several reports describe substantial disruptions to healthcare access and community infrastructure during the conflict period, and such factors have been hypothesized to influence injury risk patterns; however, these contextual elements were not directly assessed in the present study. Children’s exposure to unsafe environments—damaged playgrounds, overcrowded housing, and unregulated streets—has increased both accidental and intentional injury risks [37].

Hard flooring materials, such as marble or stone, are widely used in many urban residences in Damascus and may exacerbate the severity of facial and dental trauma following falls. Additionally, decreased parental supervision due to economic hardship, combined with limited access to emergency dental facilities, exacerbates delays in seeking care.

This context helps explain the predominance of home-based injuries (48.9%) and falls (63.7%) as etiological factors. Moreover, adolescents may face increased interpersonal violence, as reflected by a higher proportion of trauma caused by fights among children aged ≥12 years. The compounding effects of conflict, displacement, and reduced access to recreational safety infrastructure likely underline these patterns.

### 4.4. Clinical Implications

The findings have several implications for pediatric dental practice.

First, the maxillary central incisors were the most frequently injured teeth (68.5%), consistent with their anterior position and limited lip protection. This underscores the importance of early orthodontic and occlusal assessments, especially for children with increased overjet or inadequate lip coverage—conditions consistently associated with TDI risk [38].

Second, the inclusion of clinical case presentations reflects the multidisciplinary nature of pediatric dental trauma management, where orthodontic, restorative, and endodontic interventions often intersect. These cases were purposefully selected to illustrate typical injury patterns within our cohort and to demonstrate practical diagnostic and therapeutic pathways supported by adequate follow-up.

To better contextualize the advanced biomaterial discussion, it is essential to note that a considerable proportion of our sample presented late, often with pulpal or periapical complications. Such clinical realities naturally shift management from simple restorative care toward biologically oriented interventions. Modern regenerative and bioceramic-based techniques, as illustrated in our cases, offer a viable response to these delayed presentations. Their ability to promote pulp healing and root maturation—especially in immature teeth—reinforces their growing relevance in real-world pediatric trauma scenarios [39,40].

Within this context, the reference to the “biological continuum” articulated initially by Baghdadi [41] becomes especially pertinent. Rather than representing a conceptual detour, it frames the clinical progression observed in our cohort: from initial injury to biological sequelae resulting from delayed attendance to the need for biologically congruent materials capable of supporting repair. Situating our findings within the broader Preservation-to-Precision paradigm highlights a crucial clinical message: epidemiological patterns (including delays in care) directly influence the type of biomaterial-driven interventions required, thus linking population-level trends with individual-level precision care. The Preservation-to-Precision continuum referenced here is a conceptual framework previously discussed in restorative and trauma-oriented literature, in which management progresses from minimally invasive preservation of tooth structure to biologically informed, precision-based interventions. While not a formally established taxonomy in pediatric dental traumatology, it provides a useful interpretive lens for understanding how delayed presentations—such as those observed in our dataset—necessitate a shift toward regenerative and bioceramic-based therapies.

Third, SHCN children exhibited a significantly higher mean number of traumatised teeth (2.61 vs. 1.66; *p* < 0.001). This underscores the need for specialized preventive programs and targeted caregiver education focusing on motor coordination, protective devices, and supervised play.

### 4.5. Preventive and Policy Implications

The high proportion of delayed visits and home-related injuries indicates significant gaps in awareness, emergency response, and intersectoral coordination. Prevention must therefore operate at multiple levels:
School-based Programs
Introduce injury-prevention curricula that include dental first-aid education (e.g., management of avulsed teeth, proper storage media).Teachers and school nurses should receive annual first-responder training, supported by the Ministries of Education and Health.
Parental and Community Education
Launch nationwide awareness campaigns on dental trauma prevention and early care—via television, social media, and community centers.Encourage the use of mouthguards during sports activities and promote safer play environments.
Health System Strengthening
Integrate dental trauma management protocols within emergency departments and pediatric hospitals.Establish a national TDI registry and reporting system to monitor incidence, guide resource allocation, and evaluate interventions.
Urban and Infrastructure Planning
Advocate for child-safe environments through collaboration between the Ministry of Housing, local municipalities, and NGOs.Encourage the use of shock-absorbent materials (e.g., rubberized flooring) in schools and playgrounds, replacing hard marble or stone surfaces prevalent in Damascus (Syrian) homes.


Such coordinated action can significantly reduce the frequency and severity of TDIs while improving prognosis through prompt care and follow-up. As conflict-affected MENA countries experience breakdowns in infrastructure and systems, heightening children’s exposure to preventable injuries [42], rebuilding child oral-health programs must move beyond emergency repair toward resilience-based models. Aligning prevention with the *Rethink-Reform-Rise* (3R) philosophy and the *Preservation-to-Precision* continuum ensures that restorative, educational, and policy efforts evolve together to safeguard children’s health in both post-conflict and recovery settings [43]. The ‘Rethink–Reform–Rise (3R)’ philosophy, as used in this Discussion, is not an established clinical framework within pediatric dental traumatology but represents an author-adapted conceptual model intended to situate preventive and policy implications within broader post-conflict health-system rebuilding efforts. Its purpose is to emphasize strategic progression—from reassessing existing structures, to reforming care pathways, and to strengthening long-term resilience.

### 4.6. Limitations

The retrospective, hospital-based nature of this study limits its generalizability to the broader community, as it includes only children who sought care at a university clinic. Additionally, relying on recorded data may introduce reporting bias or missing variables (e.g., socioeconomic status). Despite these limitations, the large sample size, standardized WHO classification, and excellent inter-examiner reliability (κ = 0.89) ensure strong internal validity. Future prospective and community-based studies are warranted to complement these findings and assess long-term outcomes of TDI management in Syrian children. Although this study is institution-based, it is important to note that Damascus—both historically and particularly during the years of conflict—has served as a refuge and relatively safer destination for families from across Syria. As a result, the patient population at Damascus University includes children originating from multiple governorates, providing a diverse clinical cohort. Nonetheless, the findings should not be interpreted as nationally representative; any references to broader Syrian contexts in the manuscript are intended solely to situate the results within relevant public health discussions rather than to generalize empirically to the national level.

### 4.7. Summary

In summary, this study confirms that TDIs remain a significant but preventable pediatric health issue in Syria. The combination of post-conflict environmental risks, delayed care, and limited public awareness calls for urgent integration of dental trauma prevention into national child health strategies. The multidisciplinary treatment outcomes further illustrate that, with proper intervention and training, functional and aesthetic recovery can be achieved even in resource-limited settings.

## 5. Conclusions

This study provides updated institutional data on traumatic dental injuries (TDIs) among Syrian children and adolescents, revealing a prevalence of 11.08%. Males, younger children, and those with a history of previous trauma were at significantly higher risk. Falls within the home environment remained the predominant cause, while maxillary central incisors were the most frequently affected teeth. Delayed presentation—observed in nearly half of all cases—was strongly associated with pulpal and apical complications, especially asymptomatic apical periodontitis, which occurred predominantly among late presenters. This finding underscores a critical gap in emergency response and public awareness.

The results also highlight the increased burden among children with special healthcare needs, who sustained more severe and multiple injuries. These insights necessitate a multifaceted prevention approach that encompasses school safety programs, caregiver education, and structured emergency referral systems.

At the clinical level, the study’s case presentations illustrate the effectiveness of integrated, multidisciplinary management using contemporary biomaterials and regenerative protocols. Such approaches can significantly enhance prognosis, even when the initial presentation is delayed. The illustrative cases presented in this study highlight how specific disciplines contribute to comprehensive trauma care. Orthodontic intervention was most relevant in cases requiring space regaining and alignment following tooth loss; restorative procedures—including fragment reattachment and adhesive rehabilitation—were central for managing crown fractures; and endodontic or regenerative treatments were essential where pulpal or apical pathology had developed, particularly in delayed presentations. These examples underscore that multidisciplinary management is not uniform but tailored to the biological status of the injured tooth and the timing of presentation.

### Call to Action

Given the post-conflict reconstruction phase in Syria, dental trauma prevention should be embedded within broader child health and safety policies. To support practical implementation, the most immediately actionable steps include (1) establishing a national traumatic dental injury (TDI) surveillance registry to enable systematic monitoring and resource planning, and (2) integrating school-based first-aid training—particularly avulsion management and dental trauma referral protocols—into existing health and safety curricula. These measures can be deployed with relatively limited infrastructure while providing substantial public health benefits. Once these foundational components are in place, broader initiatives—such as integrating dental trauma education within national child health policies, improving urban safety design, and strengthening intersectoral coordination—can progressively build upon this framework. By integrating dental trauma management into national child welfare and educational frameworks, Syria can substantially reduce preventable injuries and advance the long-term oral health and quality of life of its youngest generation.

## Figures and Tables

**Figure 1 ijerph-22-01864-f001:**
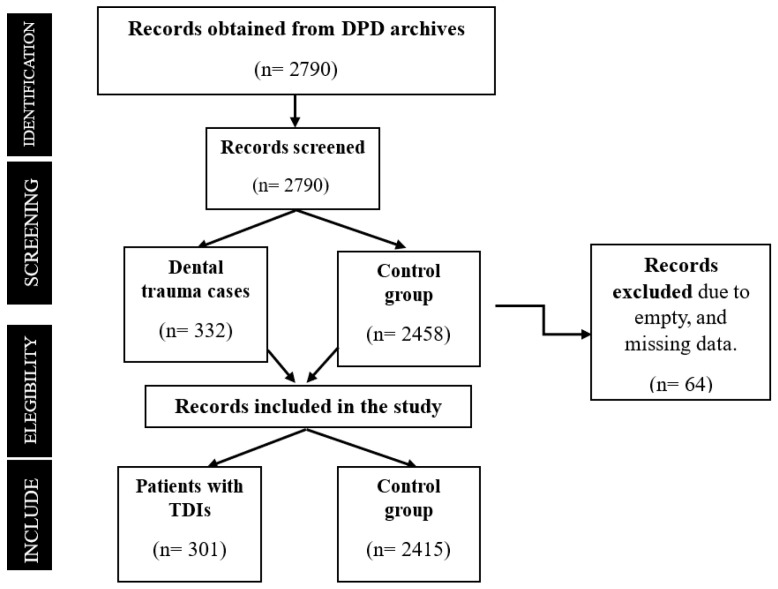
The flowchart shows the stages of record selection.

**Figure 2 ijerph-22-01864-f002:**
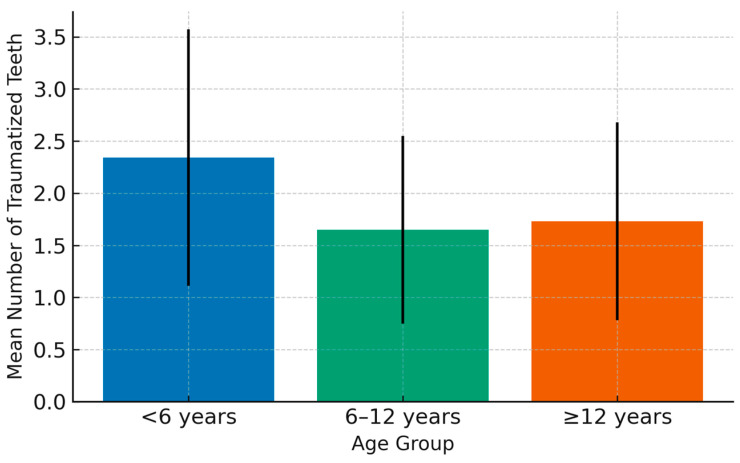
Mean (±SD) number of traumatized teeth by age group (<6 years, 6–12 years, ≥12 years). The highest mean was observed among children < 6 years (2.34 ± 1.23). Error bars represent standard deviations. *p* < 0.05 is considered significant.

**Figure 3 ijerph-22-01864-f003:**
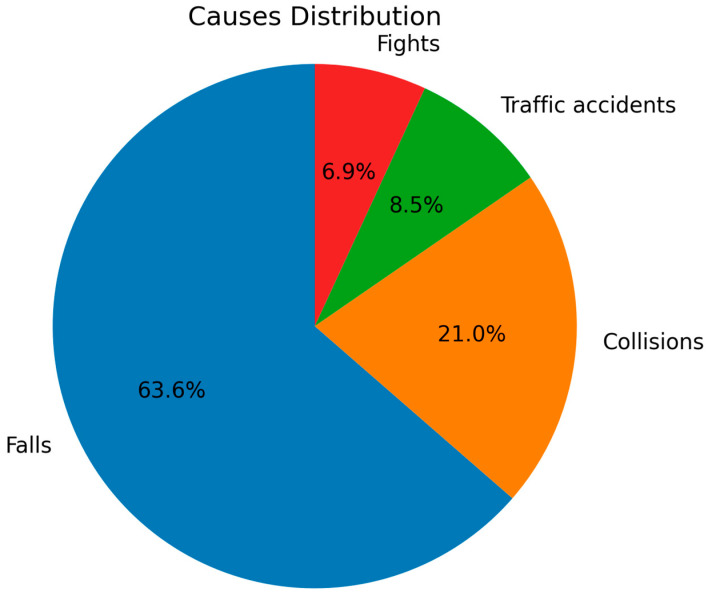
Relative distribution of etiological causes of TDIs. Falls (63.6%) were the predominant cause, followed by collisions (21.0%), traffic accidents (8.5%), and fights (6.9%). The predominance of falls reflects household environments as the primary setting of injury.

**Figure 4 ijerph-22-01864-f004:**
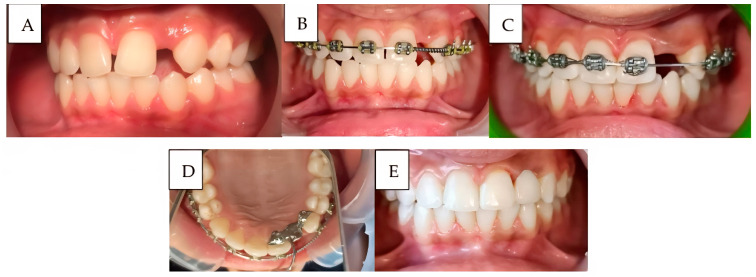
A case of anterior teeth axis correction using fixed orthodontic treatment to redistribute the space of a missing lateral incisor, followed by replacement with an adhesive bridge: (**A**): Before treatment, (**B**): After appliance placement, (**C**): Completion of orthodontic treatment, (**D**): Adhesive bridge application, and (**E**): The final result, showing the bonded bridge in place from the frontal view.

**Figure 5 ijerph-22-01864-f005:**
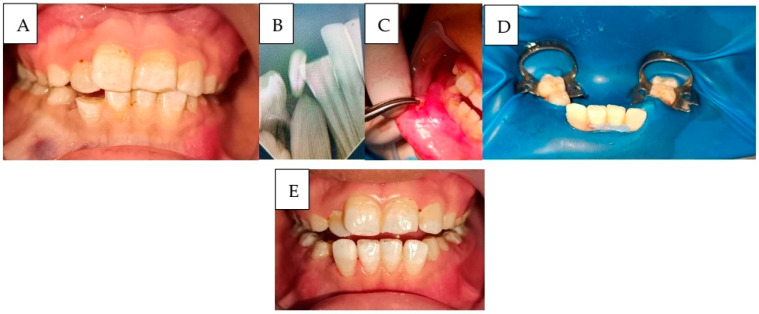
A case of managing a complicated crown fracture with the fractured fragment retained within the lower lip: (**A**): Preoperative view, (**B**): Radiograph showing the location of the broken fragment within the lower lip, (**C**): Surgical removal of the fractured fragment, (**D**): Isolation, endodontic treatment, and reattachment of the broken fragment in a single visit, and (**E**): 6 months follow-up.

**Figure 6 ijerph-22-01864-f006:**
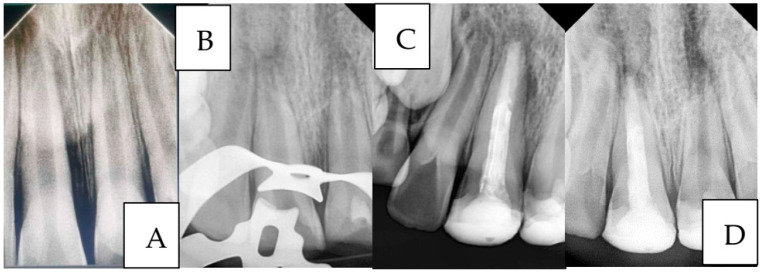
A case of managing anterior teeth with a history of repeated trauma: (**A**): Diagnostic radiograph, (**B**): Radiograph showing working length determination for tooth #11 and MTA pulpotomy for tooth #21 (**C**): Radiograph showing the placement of MTA apical plug and final restorations, and (**D**): Two-year follow-up radiograph showing successful treatment outcome of tooth #11 and failure of tooth #21.

**Table 1 ijerph-22-01864-t001:** Demographic and behavioural characteristics associated with traumatic dental injuries (TDIs).

Variable	With Injuries (n = 301)	No Injuries (n = 2415)	TDI Prevalence (%)	*p*-Value
Gender (Male/Female)	213/88	1323/1092	13.9/7.5	<0.001
Age group (<6/6–12/≥12)	32/165/104	441/1507/467	6.8/8.4/18.2	<0.001
Health status (Healthy/SHCN)	273/28	2169/246	11.8/10.2	0.631
Frankl’s behaviour (Neg/Pos)	104/197	1125/1290	8.1/17.6	<0.001
Previous trauma (Yes/No)	17/284	53/2354	24.3/10.8	<0.001
Occlusion (Class I/II/III)	212/53/6	1447/309/146	12.8/14.6/4.0	<0.001

Note: SHCN = special healthcare needs.

**Table 2 ijerph-22-01864-t002:** Binary logistic regression analysis for predictors of traumatic dental injuries.

Predictor Variable	B	SE	Wald	OR (95% CI)	*p*-Value
Gender (Male)	0.834	0.137	37.262	2.30 (1.76–3.01)	<0.001
Age (Increasing)	−0.570	0.138	17.064	0.56 (0.43–0.74)	<0.001
Behaviour (Frankl)	−0.140	0.090	2.415	0.87 (0.73–1.04)	0.120
Previous trauma	0.017	0.120	0.019	1.02 (0.80–1.29)	0.889
Angle’s classification	0.095	0.059	2.574	1.10 (0.98–1.23)	0.109

Model χ^2^ = 62.45, *p* < 0.001; Nagelkerke R^2^ = 0.118; Hosmer–Lemeshow test *p* = 0.47; Overall classification accuracy = 78.2%. OR = odds ratio; CI = confidence interval.

**Table 3 ijerph-22-01864-t003:** Comparison of mean caries indices (DMFT/dmft) between children with and without TDIs.

Index	With Injuries (Mean ± SD)	No Injuries (Mean ± SD)	Mean Difference (95% CI)	*p*-Value
dmft (Primary)	1.96 ± 2.15	5.92 ± 3.79	−3.96 (−4.35, −3.57)	<0.001
DMFT (Permanent)	2.94 ± 2.33	5.57 ± 4.00	−2.63 (−3.05, −2.21)	<0.001
DMFT + dmft (Mixed)	4.45 ± 2.82	5.51 ± 3.44	−1.06 (−1.45, −0.67)	<0.001

**Table 4 ijerph-22-01864-t004:** Injury characteristics, pulp diagnosis, and treatment modalities among permanent and primary teeth (n = 521).

Variable	Permanent Teeth (n = 430)	Primary Teeth (n = 91)	Notes/*p*-Value
Apex status—open	40.1%	–	40.1% overall open apex rate
Most affected teeth	#11 (36.1%), #21 (32.4%)	#51 (5.4%), #61 (5.0%)	—
Complicated crown fracture	39.1%	9.9%	Most frequent type overall
Avulsion	4.9%	25.3%	*p* < 0.001 (primary > permanent)
Healthy/Reversible/Irreversible pulpitis	22.0%/29.6%/2.7%	16.4%/6.0%/9%	—
Symptomatic/Asymptomatic apical periodontitis	13.6%/32.1%	50.7%/17.9%	Delayed presentation (>14 weeks) was more frequently linked to teeth diagnosed with asymptomatic apical periodontitis (χ^2^ = 18.7, *p* < 0.001)
>4 weeks delay between the TDI and the treatment	45.3%	45.1%	45% delayed overall, no significant differences were observed between SHCN and healthy children regarding the time between trauma and dental visit (χ^2^ = 4.1, *p* = 0.257)
Restorations	26.1%	1.9%	
Re-bonding the fractured segment	1.2%	—	
Traditional root canal treatment	7.7%	1.7%	
Apexogenesis	7.8%	0.3%	
Apical barrier, Regeneration	18.3%, 2.7%	—	—

Note: Percentages refer to teeth within each dentition group.

**Table 5 ijerph-22-01864-t005:** Statistical tests to study the differences between the following variables: gender, health status, previous traumatic injury, cause, and age classification in children with TDIs.

Variables		Age Classification					*p*-Value
			<6	6-<12	12≤	Total	
Gender	Male	N	15	121	77	213	0.007 ^a^
		% of Total	5.0%	40.2%	25.6%	70.8%	
	Female	N	17	44	27	88	
		% of Total	5.6%	14.6%	9.0%	29.2%	
	Total	N	32	165	104	301	
		% of Total	10.6%	54.8%	34.6%	100.0%	
Health status	Healthy	N	22	157	94	273	<0.001 ^a^
		% of Total	7.3%	52.2%	31.2%	90.7%	
	SHCN	N	10	8	10	28	
		% of Total	3.3%	2.7%	3.3%	9.3%	
	Total	N	32	165	104	301	
		% of Total	10.6%	54.8%	34.6%	100.0%	
Previous traumatic injury	No	N	29	160	95	284	0.094 ^a^
		% of Total	9.6%	53.2%	31.6%	94.4%	
	Yes	N	3	5	9	17	
		% of Total	1.0%	1.7%	3.0%	5.6%	
	Total	N	32	165	104	301	
		% of Total	10.6%	54.8%	34.6%	100.0%	
Cause	Falls	N	20	130	60	210	<0.001 ^a^
		% of Total	6.7%	43.3%	20.0%	70.0%	
	Traffic accident	N	5	5	5	15	
		% of Total	1.7%	1.7%	1.7%	5.0%	
	Collisions with objects	N	6	27	22	55	
		% of Total	2.0%	9.0%	7.3%	18.3%	
	Fight	N	1	2	17	20	
		% of Total	0.3%	0.7%	5.7%	6.7%	
	Total	N	32	164	104	300	
		% of Total	10.7%	54.7%	34.7%	100.0%	

a. Chi-Square.

**Table 6 ijerph-22-01864-t006:** Statistical tests to study the differences between the following variables: gender, health status, occlusion type, place, and age.

Variables		Number of Traumatized Teeth			*p*-Value
		N	Mean	SD	
Gender	Male	213	1.71	0.955	0.305 ^a^
	Female	88	1.84	1.027	
Health status	Healthy	273	1.66	0.918	<0.001 ^a^
	SHCN	28	2.61	1.133	
Place	Home	148	1.74	0.897	0.469 ^b^
	Street	46	1.98	1.483	
	School	62	1.65	0.812	
	Sport	19	1.79	0.855	
	Garden	25	1.64	0.700	
Total		300	1.75	0.978	
Age	<6	32	2.34	1.234	0.001 ^b^
	6–<12	165	1.65	0.903	
	12≤	104	1.73	0.947	
Total		301	1.75	0.977	
Cause	Falls	210	1.59	0.760	<0.001 ^b^
	Traffic accident	15	2.93	1.944	
	Collision with objects	55	2.04	1.105	
	Fight	20	1.80	0.894	
Total		300	1.75	0.978	

a. Independent samples test. b. One-Way ANOVA.

**Table 7 ijerph-22-01864-t007:** Post hoc tests for significant differences between the mean number of traumatized teeth and the variables: occlusion type, age, and trauma cause.

Age		Mean Difference	Std. Error	*p*-Value ^a^
<6	6–<12	0.695	0.185	0.001
	12≤	0.613	0.194	0.005
6-<12	12≤	−0.082	0.120	1.000
Cause		Mean Difference	Std. Error	*p*-value ^a^
Falls	Traffic accident	−1.343	0.248	0.000
	Collisions with objects	−0.446	0.141	0.010
	Fight	−0.210	0.217	1.000
Traffic accident	Collisions with objects	0.897	0.270	0.006
	Fight	1.133	0.317	0.002
Collisions with objects	Fight	0.236	0.242	1.000

a. Post hoc tests (Bonferroni).

## Data Availability

The data are available upon request from the authors.

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
