# Peer review of "Epidemiology Meets Advocacy: Understanding Pediatric Dental Trauma and Delayed Care in Post-Conflict Syria"

_ijerph, 2025, doi:10.3390/ijerph22121864_

Round 1
Reviewer 1 Report
Comments and Suggestions for Authors
Thank you for the opportunity to review this manuscript. Both the topic and the context are very interesting with the capacity to improve to sound publication in the field. However, I have several concerns related to the current version of the manuscript. They are as follows:
- Section 2.2. In the inclusion criteria it should be clarified if the patient were included only if the TDI was the cause of the referral to the hospital, or it was concomitant condition for general trauma, and also were the cases of old (previous) trauma that had been diagnosed during routine clinical examination were included in the study population
- Section 2.4. Definition of delayed treatment (in hours and/or days) should be clearly defined, since it is mentioned in the title and also present very important results that could improve interpretative potential of the whole research
- Section 3.3 Time between the injury and the treatment should be clearly stated. Also, the findings of the data regarding if the some of the patients received first aid and after that referred to the hospital would give additional valuable information
- Section 3.5. The reasons for choosing illustrative cases should be elaborated
- Table 6. The place of injury was classified into home, street, school, sport and garden. Based on what methodological criteria?
Author Response
As per attached file.

Reviewer 2 Report
Comments and Suggestions for Authors
The manuscript entitled “Epidemiology Meets Advocacy: Understanding Pediatric Dental Trauma and Delayed Care in Post-Conflict Syria” presents a retrospective cross-sectional analysis of pediatric dental trauma cases at Damascus University during 2023–2024. The topic is relevant, timely, and valuable, particularly in the context of post-conflict public health challenges. The large sample size (n = 2,716), structured methodology, and integration of illustrative clinical cases provide depth and practical relevance.
Overall, the article is well written, scientifically grounded, and of potential interest to the journal readership. Several areas, however, require clarification, strengthening, or streamlining to enhance methodological transparency and interpretability
1) While the hospital-based design is acknowledged as a limitation, the manuscript at times generalizes findings to the national level. Given the institutional nature of the sample, generalization beyond urban or university-referred populations should be moderated or supported by additional context.
2) A major finding is that 45.3% of children presented more than one month after injury (Table 4, page 6). This is an important observation, but the manuscript does not sufficiently explore: specific barriers (financial, logistical, awareness-related), how these delays compare to pre-conflict patterns, or whether delays differ between SHCN and healthy children.
3) The three detailed case presentations are informative, but they dominate sections of the results and discussion. For an epidemiological study, the narrative should prioritize aggregated data; the cases could be shortened or moved to an appendix unless the journal encourages such inclusions.
4) Several statistical results require clarification: The logistic regression model explains only 11.8% of the variance (Nagelkerke R² = 0.118, page 5). This should be explicitly discussed as evidence of unmeasured confounding. Behavioural variables (Frankl scale) showed significance in univariate analysis but not in logistic regression. This deserves comment regarding collinearity or age/gender influence. Some p-values are given without effect sizes; for clinical relevance, effect sizes are essential.
5) The discussion introduces an extended conceptual framework linking biological healing, biomaterials, and personalized care. This is an interesting and modern approach, but the transition from epidemiology to advanced biomaterial science feels abrupt. Consider tightening this narrative or more clearly linking it to observed outcomes.
Author Response
As per attached file.

Reviewer 3 Report
Comments and Suggestions for Authors
Dear authors,
Please see the attachment.

Author Response
As per attached file
